# Inactivating the Uninhibited: The Tale of Activins and Inhibins in Pulmonary Arterial Hypertension

**DOI:** 10.3390/ijms24043332

**Published:** 2023-02-07

**Authors:** Gusty Rizky Teguh Ryanto, Ahmad Musthafa, Tetsuya Hara, Noriaki Emoto

**Affiliations:** 1Laboratory of Clinical Pharmaceutical Science, Kobe Pharmaceutical University, Kobe 658-8558, Japan; 2Division of Cardiovascular Medicine, Department of Internal Medicine, Kobe University Graduate School of Medicine, Kobe 650-0017, Japan

**Keywords:** pulmonary arterial hypertension, vascular remodeling, activin

## Abstract

Advances in technology and biomedical knowledge have led to the effective diagnosis and treatment of an increasing number of rare diseases. Pulmonary arterial hypertension (PAH) is a rare disorder of the pulmonary vasculature that is associated with high mortality and morbidity rates. Although significant progress has been made in understanding PAH and its diagnosis and treatment, numerous unanswered questions remain regarding pulmonary vascular remodeling, a major factor contributing to the increase in pulmonary arterial pressure. Here, we discuss the role of activins and inhibins, both of which belong to the TGF-β superfamily, in PAH development. We examine how these relate to signaling pathways implicated in PAH pathogenesis. Furthermore, we discuss how activin/inhibin-targeting drugs, particularly sotatercep, affect pathophysiology, as these target the afore-mentioned specific pathway. We highlight activin/inhibin signaling as a critical mediator of PAH development that is to be targeted for therapeutic gain, potentially improving patient outcomes in the future.

## 1. Introduction

Pulmonary arterial hypertension (PAH) is a rare condition characterized by the progressive remodeling of arteries, especially the precapillary region, which increases pulmonary artery pressure, eventually culminating in right heart failure [1,2]. PAH is a subtype of pulmonary hypertension (PH). Half a century ago, this condition was a complete mystery to physicians, resulting in high mortality and morbidity. Our current understanding of PAH has significantly improved the survival rate of patients to approximately 90%. To date, three different pathways can be targeted using different therapeutic agents, with good therapeutic efficacy [1,3]. Nevertheless, mortality among intermediate- and high-risk PAH patients remains high, highlighting the need for better disease management [4]. Novel treatment options are to be proposed based on a deeper understanding of PAH pathogenesis [5,6,7]. Here, we focus on the activin/inhibin family of molecules, which have recently gained traction as promising therapeutic targets for PAH. Our discussion will focus on how activin/inhibin may influence vascular remodeling in PAH. Furthermore, we will discuss the current clinical landscape of drugs targeting these factors.

## 2. Current Understanding of PAH Pathogenesis

PAH is a form of pre-capillary PH arising due to the primary elevation of pressure within the pulmonary artery system without the presence of chronic lung disease, which causes hypoxemia or pulmonary artery obstruction. The most recent hemodynamic criteria for pre-capillary PH as per the 2022 European Society of Cardiology and European Respiratory Society Guidelines for the diagnosis and treatment of PH are defined as an increase in mean pulmonary artery pressure (mPAP) greater than 20 mmHg at rest, pulmonary vascular resistance (PVR) greater than 2 Wood units, and pulmonary capillary wedge pressure of no more than 15 mmHg [1]. PAH is one of the five clinical subtypes of PH, which are grouped based on similarities in pathophysiology, clinical and hemodynamic profile, and management. PAH is further differentiated into several subsets. The most recent classification is depicted in Table 1 [1]. Among the various diagnoses that are considered PAH, common pathological features are apparent, with remodeling and abnormalities of all three pulmonary vessel layers [2,8].

The vascular intimal layer consists of endothelial cells (ECs) that line the luminal surface of vessels. Healthy endothelial cells maintain homeostasis and regulate vascular tone in response to diverse stimuli. During PAH, various endothelial cell abnormalities are observed that greatly contribute to the remodeling of pulmonary vasculature [7]. The proliferation of lung vascular endothelial cells (LVECs) is increased in patients with idiopathic pulmonary arterial hypertension (IPAH) [9]. Similarly, in bovine pulmonary arterial endothelial cells (PAECs), the ROCK2 pathway is upregulated under hypoxic conditions, promoting proliferation [10]. A significant increase in senescence-associated secretory phenotype (SASP)-related gene expression is noted in the monocrotaline PAH model. Hemodynamic unloading does not reverse this state, suggestive of different severity levels resulting from different mechanisms [11]. Inflammation plays a major role in the vascular cell dysfunction observed in patients with PAH. In particular, endothelial cell pyroptosis, an inflammatory form of programmed cell death, is noted to occur during PAH. In rats with monocrotaline-induced PH, endothelial cells undergo pyroptosis induced by caspase-11 activation of gasdermin D and gasdermin E [12]. In systemic lupus erythematosus patients with PAH, lipopolysaccharide (LPS)-induced pyroptosis contributes to defective BMPRII (bone morphogenetic protein receptor type 2) signaling, which has a central role in PAH pathogenesis [13]. LPS is a component of the bacterial cell wall that triggers activation of toll-like receptors (TLRs), mainly TLR4, and the subsequent activation of pro-inflammatory signaling pathways, such as NF-κB, which drives inflammatory cytokine expression (e.g., IL-6), with many cytokines participating in the pulmonary vascular remodeling through perivascular inflammation [14]. An increase in serum LPS level, thought to be related to the gut dysbiosis commonly observed in PAH patients, has been noted in preclinical animal models of PAH, such as monocrotaline-treated rats, as well as in various forms of PAH, including IPAH, HPAH, PAH, and the aforementioned PAH due to SLE [13,14,15].

The dysfunctional ECs of PAH patients undergo endothelial-to-mesenchymal transition (EndMT). LVECs isolated from IPAH patients exhibit increased expression of SNAI1, SNAI2, and smooth muscle cells (SMCs) lineage markers, including ACTA2, FSP1, and FN1 [9]. The pulmonary arteries of healthy patients without PAH have a thin layer of ECs with diffuse expression of endothelial markers (CD34, CD31, and VE-cadherin), adjacent to SMCs that express alpha-smooth muscle actin (alpha-SMA). Nevertheless, there is evidence of alpha-SMA expression in the intimal and luminal regions of plexiform lesions, as well as in the subendothelial cells of patients with PAH. These cells expressed both endothelial and mesenchymal markers [16]. The EndMT can be mediated via hypoxia signaling pathways, such as PHD2/HIF2α. PHD2 knockout in ECs triggers EndMT and induces PH under normoxic conditions, whereas HIF2α-knockout mice are protected against PH under hypoxic conditions [9]. The proliferation and migration capacities of neighboring endothelial and mesenchymal cells are enhanced in the presence of ECs undergoing EndMT [17]. The aforementioned abnormal growth of ECs and/or medial SMCs may result in the development of plexiform lesions [18,19].

The second blood vessel layer, the medial layer, consists of SMCs, and various changes can be observed in the PASMCs of PAH vessels. In response to diverse stimuli, mostly from ECs, it can contract or relax, modifying the lumen diameter of blood vessels and altering vascular resistance. Normally, ECs produce prostacyclin, a potent platelet aggregation inhibitor and vasodilator [5]. However, the low expression of its cognate receptor was noted in PASMCs isolated from patients with IPAH compared to that in normal controls, indicative of a potential reduction in active prostacyclin signaling. In addition to inducing vasodilation via cAMP generation, prostacyclin can inhibit the proliferation of PASMCs [20]. In patients with primary and secondary PH, thromboxane A2 release was elevated while prostacyclin release was decreased [21]. These observations support the use of prostacyclin analogs in clinical PAH. Another vasodilatory signaling factor, nitric oxide (NO), affects SMCs through cGMP activation and its downstream signaling. The loss of this signaling axis could lead to uncontrolled SMCs proliferation and altered apoptotic potential, which is why this pathway is another target for PAH therapy via phosphodiesterase V (PDE-V) inhibition or the administration of soluble guanyl cyclase (sGC) to induce cGMP activation [22]. A third group of EC-produced peptides acting on SMCs are endothelins (ETs), specifically endothelin-1 (ET-1). ET-1 is a potent vasoconstrictor signaling through endothelin receptor A (ET_A_) and B (ET_B_) in SMCs (and also ECs) and known to be upregulated in various PAH etiologies. The use of endothelin receptor antagonists to block this signaling has become routine in PAH treatment [23].

Beyond the three commonly targeted pathways that modulate SMCs, other signaling axes are also dysregulated. For example, in SU5416/hypoxia model rats, Kv11.1, a voltage-gated potassium channel, is upregulated in the small arteries [24]. An increase in transient receptor potential cation (TRPC) enhances both store-operated and receptor-operated calcium entry [25]. These factors contribute to an increase in vascular tone. Similar to endothelial cells, PASMCs isolated from patients with IPAH proliferated faster and initially had longer telomeres than control cells. In addition, IPAH-PASMCs had lower Bax and elevated Bcl2 protein levels, indicative of apoptotic resistance [26]. A small number of SMCs with high Notch3 expression penetrated the internal elastic lamina, resulting in neointimal lesions, severely narrowing the vessel lumen in a house dust mite PAH mice model [18]. Mitochondrial dysfunctions in ECs and SMCs have been implicated as a cause of the metabolic phenotype that is increasingly observed in PAH patients with different etiologies [27]. The metabolic dysregulation caused by mitochondrial dysfunction in the vasculature includes aerobic glycolysis (also known as the Warburg Effect), decreased fatty acid oxidation, insulin resistance, and alterations in the lipid profile of patients [28]. Indeed, a study showed how PAH patients have increased lipid and lipoprotein levels in addition to glucose intolerance and insulin resistance [29]. All of these metabolic changes contribute to EC and SMC dysfunction, thus acting as a major driver of vascular remodeling.

Despite being furthest from the lumen, the adventitial layer also plays a role in PH. In addition to increased intimal and medial thicknesses, adventitial thickening was observed in patients with primary PH [30]. In monocrotaline-injected rats, knocking out the *TRPV4* gene reduced adventitia and media hypertrophy by blunting adventitial remodeling [31]. Microarray analysis has revealed that adventitial layer-residing immune cells of the lung from IPAH patients have distinct profiles compared to those of control patients [32]. The upregulation of IL-6 following hypoxia triggers M2 polarization of adventitia-residing macrophages in mice [33]. Furthermore, conditioned medium from M2-polarized macrophages induced the proliferation of PASMCs, a critical aspect of vascular remodeling in PAH [34].

Although a great number of overlapping and related pathways are affected by different PAH causes, molecular, functional, and histological differences do exist between PAH types and should be noted. In patients with PAH due to congenital heart disease, most are children and have different histological characteristics compared to IPAH patients. Children with PAH will develop greater medial hypertrophy and less plexiform lesions while the opposite is observed in adults with PAH [35]. Patients with PAH due to scleroderma exhibit a worse right ventricular phenotype compared to IPAH patients, most likely due to the heavy involvement of sarcomeres in scleroderma-PAH [36]. Portopulmonary hypertension, although having similar molecular signatures with IPAH, is associated with more favorable hemodynamic features, yet it has a worse prognosis [37]. One common thread among the etiologies, however, is the dysfunction in BMPRII signaling, induced through various mechanisms. In the majority of heritable PAH and some non-heritable PAH, germline mutations of the gene encoding BMPRII, member of the transforming growth factor-β (TGF-β) superfamily, have been implicated as contributing factors. A heterozygous mutation was found to impair its function by causing premature truncation, impairing the formation of heteromeric receptor complexes, affecting ligand binding, and altering signal transduction [38,39,40,41,42]. Through the inhibition of Smad-1 and p38/MAPK, reduced bone morphogenetic protein (BMP) signaling resulted in greater proliferation of hPASMCs, one of the major causes of PAH. Thus, phospho-Smad1 was downregulated in the intimal and medial cells of PAH lungs when compared to levels in control lungs [43]. A heterozygous BMPRII mutation (BMPRII +/−) in mice leads to higher right ventricular systolic pressure (RVSP) as a consequence of increased thromboxane production, followed by muscular thickening of the pulmonary vessel under inflammatory stress [44]. Furthermore, rats with heterogeneous deletions in the first BMPRII exon developed PH associated with vascular remodeling, mediated by the overexpression of Twist and P-vim [16]. In PAECs, BMPRII silencing also promotes EndMT by upregulating the EndMT transcription factors Snail and Slug, but not Twist, owing to an increase in the expression of the chromatin architectural factor high mobility group AT-hook1 (HMGA1) [45]. The BMP pro-apoptotic effect is mediated via the activation of caspase-9 and caspase-8 by BMPRII. Thus, reductions in BMPRII levels or activity due to mutations can result in resistance to apoptosis and subsequent vascular SMC hyperproliferation [46]. Multiple studies have reported the importance of BMP9/10 ligands in regulating lung vasculature, particularly ECs and SMCs. In particular, restoring BMP9/10 signaling may have a positive effect in PH animal models [47,48,49]. The downregulation of BMPRII signaling is not limited to IPAH and HPAH, with other etiological causes of PAH linked to reductions in BMPRII and its downstream signaling [50]. Taken together, BMPRII signaling defects may contribute to PAH through multiple mechanisms.

## 3. Activins and Inhibins

As mentioned in the previous section, BMPRII signaling dysregulation has been established across PAH of various etiologies. Other members of the 33-molecule TGF-β superfamily have also been implicated in PAH pathogenesis. This includes all isoforms of TGF-β, GDFs (growth differentiation factors), and other BMPs [51]. It is therefore plausible that activins and inhibins, as members of the same superfamily, may be implicated in PAH pathogenesis.

Activins and inhibins are glycoproteins derived from the dimerization of INH family members, which include INHBA, INHBB, INHBC, INHBE, and INHA [52,53]. Activins and inhibins have been extensively studied in a range of different organs [52]. Following its discovery in the 1920s, inhibin was considered a reproductive hormone until activin was later discovered in the 1980s [54]. Both proteins have been purified and characterized [52]. The name activin was given because of its ability to stimulate the release of follicle-stimulating hormone (FSH) from the pituitary gland [53]. Activin and inhibin production involves the transcription of INH chain mRNA, which is then transmuted into precursor chains and undergoes dimerization to form precursor proteins [52,53]. Activin/inhibin pre-proteins are then cleaved by furin-like proteases, folded, and reassembled into their correct conformation. A biologically active form of activin/inhibin is formed as a result. The synthesis of the active form depends on the combination of dimerized INH family chains. As an example, activin A is produced by the homodimerization of two INHBA chains linked via disulfide bonds [55]. In contrast, inhibin A is produced when an INHBA chain is dimerized with an INHA chain [54]. Table 2 lists the currently known isoforms of the activin/inhibin family. Notably, activin dimers are sometimes also referred to as including β in their names (e.g., activin A is sometimes also called activin βA). To avoid confusion, we will use the term without including β (e.g., activin A) in this manuscript.

Since their discovery, activin/inhibin expression and function have been characterized, allowing further insight into their molecular functions. First, a significant body of evidence suggests that activins signal through the canonical activin receptors type IIA and IIB (ActRIIA and ActRIIB), which are serine/threonine kinases. As with the other receptors of the TGF-β superfamily, activin receptors form complexes with their type I receptor after ligand binding, predominantly activin-like kinase 4 (ALK4), also known as activin receptor type IB (ActRIB), or ALK7, also known as activin receptor type IC (ActRIC) [52,53]. With the aid of Smad anchor for receptor activation (SARA), activins induce canonical signaling when bound by phosphorylating intracellular Smads, in this case Smad-2 and Smad-3. Furthermore, these are targeted by a co-Smad, Smad-4, which then translocates to the nucleus to induce transcription [56]. Naturally, target genes differ between activin/inhibin isoforms and receptors. Considering the promiscuity of TGF-β superfamily ligands and receptors with varying degrees of affinity, other ligands, including BMPs, nodal, and GDFs, may bind to activin receptors, activating various signaling and subsequent physiological responses [56,57]. Activin/inhibin can bind to non-canonical receptors other than ActRII-ActRI [57]. There is evidence that activin B can bind to ActRIIB or ActRIIA with non-canonical type I receptors known as ALK2 or ALK3, also known as the BMP type I receptor [58,59]. As we later discuss, activin A can also bind to non-canonical receptors. In addition to non-canonical activin signaling, a number of common pathways have been associated with the activin family, including, but not limited to, MAPK/ERKs, Wnt/β-catenin, Akt/PI3K, and Tak1/p38 [60]. As a consequence, activins mediate diverse signaling cascades.

As their name implies, inhibins suppress activin-driven signaling through competitively binding receptors [54,61]. Among the two inhibin isoforms, inhibin A was first discovered in the 1920s [52,62]. As activin inhibitors, they bind to similar receptors, that is, ActRIIs in a complex with the type III TGF-β receptor betaglycan. Consequently, downstream Smad signaling via the canonical receptor is impeded [52]. Inhibins may also activate distinct signaling pathways on their own, yet further studies are needed to confirm this notion [63]. Other molecules can also interact with or inhibit activins and inhibins. These include follistatin, a secreted protein that can directly bind to activins and neutralize them, thus preventing receptor binding. A transmembrane glycoprotein called bone morphogenic and activin membrane-bound inhibitor (BAMBI) suppresses BMP and activin signaling, and the same is true for Smad-7, which is the intracellular antagonist of Smads [64,65,66].

Activin and inhibin signaling has a major impact on physiology and pathophysiology, which was already evident upon the recognition of its role in FSH stimulation and overall gonadal physiology [52]. Until recently, no molecular mechanisms had been put forth to explain the function of activins other than activin A, B, and AB. The other active isoforms (activin C and activin E) have been identified and isolated but have not been clearly implicated in physiological functions. In recent years, activin C has been shown to signal through the canonical ActRII-Smad-2/3 axis, utilizing ALK7 as its type I receptor, whereas activin E appears to be involved in energy and adipocyte homeostasis [67,68]. In terms of physiological roles, activin A is the most widely studied among isoforms. Activin A promotes the proliferation, apoptosis, and migration of cells [53]. Activin A is implicated in several diseases, including various forms of cancer, muscle dystrophy, asthma, atherosclerosis, heart failure, as well as in skin wound healing [69,70,71,72,73,74,75]. In cancer, activin A has been shown to regulate tumor cell proliferation and migration while in the skin, it may influence scar tissue formation by fibroblasts [72,76]. The presence of activin A is also known to negatively regulate muscle mass, which is of major relevance in muscle dystrophy [77]. Aside from activin A involvement in mesothelioma or oral cancers, activin B has been implicated in adipogenesis and pancreatic β-cells through its signaling via ALK7/ActRIC, another recently established adipocyte marker [78,79,80]. Activin B is also implicated in inflammation, where non-canonical Smad-1/5/8 signaling activation induces a pro-inflammatory phenotype in hepatocytes [59]. In comparison with the other two isoforms, activin AB is relatively understudied. Inhibin A and inhibin B have been suggested to modulate progesterone production, similar to activin A and B [81]. Both inhibins have been extensively studied in the context of ovarian function and fertility. The significance of their levels in serum is currently being explored in the clinical setting [48]. Like activins, inhibins are also implicated in various forms of cancer, thus representing potential therapeutic targets [62].

## 4. Roles of Activin/Inhibin in PAH Development

As stated, the TGF-β superfamily is thought to be heavily involved in PAH pathophysiology [82]. With the breakthrough discovery of a PAH-causing BMPRII mutation, the last two decades have brought about considerable advances in our understanding of PAH molecular pathogenesis. These include the discoveries of TGF-β superfamily members and their receptors, including, but not limited to, BMPs, TGF-β, ALKs, and related factors, such as endoglin (ENG) [83]. Furthermore, the similarities in downstream signaling and the ability of ligands from this family to bind different receptors suggest that other TGF-β members may be implicated in PAH [83]. As such, activin/inhibin may contribute to PAH pathogenesis. An experiment conducted in 2011 found an increase in activin A levels in the PASMCs of patients and experimental monocrotaline-induced rats [84]. Over the past few years, the involvement of activin A in PAH has been clearly established. It appears that there could be multiple sources of activin A in the pulmonary artery. INHBA/activin A was reported as highly expressed in PAECs, and others have emphasized the importance of PASMC-derived activin A, which we will discuss in this manuscript [85,86].

We previously reported that EC-derived activin A could act in an autocrine manner to bind and accelerate BMPRII degradation via lysosomes [85]. Indeed, members of the TGF-β superfamily exhibit varying affinities for multiple receptors [56,57]. Thus, activin A can bind BMPRII, in addition to its canonical receptors. By overexpressing INHBA chains or treating endothelial cells with activin A, we demonstrated that high activin A levels could reduce BMPRII. Consequently, this results in the accelerated degradation and downregulation of the canonical BMP/BMPRII/Smad-1/5/8 signaling pathway, which is essential for maintaining EC homeostasis [85]. As discussed in the previous section, BMP, especially BMP9/10, is a vital mediator of normal EC physiology, and the rescue of BMP9 signaling could reverse the PH phenotype [48].

Our study demonstrated that activin A modulates angiogenesis in ECs [85]. As other reports have suggested, activin A acts as an anti-angiogenic factor. We and others have shown that ECs derived from various vascular beds inhibit neovascularization through a VEGF-dependent pathway. Such vascular beds include pulmonary artery endothelial cells (PAECs), human microvascular endothelial cells (HMVECs), human retinal endothelial cells (HREC), human umbilical vein endothelial cells (HUVECs), recombinant activin A administration, or INHBA overexpression [85,87,88]. The anti-angiogenic capability could be, at least in part, related to pro-apoptotic capacity, as shown in various studies [88,89]. Again, while multiple signaling pathways may play a role in the pro-apoptotic activity of activin A, loss of the canonical BMP-triggered pathway is one candidate [89,90,91]. Additionally, activin A exhibits this pro-apoptotic effect not only in ECs. The importance of angiogenesis in PAH pathogenesis further supports a role for activin A in the condition [92].

Another proposed mechanism for activin A involves a different cellular source, namely, SMCs, which have also been identified as a source of INHBA/activin A expression. Yung et al. found that inhibiting activin A restored the aberrant TGF-β signaling commonly observed in PAH [51,86]. By inhibiting activin A with its ligand trap, sotatercept, they were able to tilt TGF-β signaling back towards the physiological BMP/BMPRII/Smad-1/5/8 axis in SMCs. Furthermore, sotatercept treatment could restore BMP9 levels [86]. In PASMCs, activin A acts through canonical ActRII/Smad-2/3 signaling pathway, which can induce the expression of two proteins known to inhibit BMP activity, gremlin and noggin [86,93]. Indirect inhibition of BMP signaling is believed to result in an imbalance of TGF-β superfamily signaling. Accordingly, activin A may lead to an increased proliferative capacity and an altered expression of myogenic as well as fibrogenic genes, both of which are hallmarks of vascular remodeling in PAH. Finally, vascular inflammation has been implicated as a major cause of PAH for decades. In this regard, Joshi et al. found that the inhibition of activin A with its ligand trap analog RAP-011, whether alone or in combination with the vasodilator sildenafil, inhibited the pro-inflammatory phenotype observed in the vasculature of SU5416/Hypoxia rats. This phenomenon was not observed in rats treated with vehicle or sildenafil alone [94]. Activin A contributed to macrophage activation into a pro-inflammatory phenotype, resulting in lower macrophage infiltration in lung tissue [94]. The macrophage activation observed in activin A-treated THP-1 cells is notably similar to LPS stimulation in terms of the upregulated pro-inflammatory cytokines important in PAH pathogenesis (e.g., IL-1β, IL-6, and TNF-α) [95]. Figure 1 illustrates the proposed mechanism of action of activin A in PAH pathophysiology.

We and others have demonstrated the involvement of activin A/INHBA in animal models of PAH. Our group confirmed that the EC-specific overexpression of INHBA could induce a spontaneous PH that was exacerbated by chronic hypoxia. According to another group, administration of sotatercept improves the hemodynamics of various established PH models, including the SU5416/Hypoxia rat model. This has been shown to result in severe vascular remodeling [85,86,94]. We knocked out the INHBA gene in mouse ECs and found that knockout mice exhibited an amelioration of the chronic hypoxia-induced PH phenotype compared to control wild-type mice. This suggests that INHBA/activin A influences PAH development [85]. Joshi et al. confirmed that inhibiting activin A, in addition to GDF8 and GDF11, through ligand trap administration could suppress inflammation in various in vivo models of PH [94].

Clinically, several studies have shown the direct involvement of activin and inhibin in patients with PAH, most commonly those with IPAH and HPAH. We discovered that ECs isolated from IPAH patients expressed higher levels of INHBA. In addition, they secreted more activin A than ECs isolated from normal donor lungs. In contrast, Yung et al. showed that an increase in activin A-positive cells could be found in the lung vasculature of PAH patients [85,86]. Furthermore, they found an increase in serum activin A levels in PAH patients, which is in line with the results of a 2011 study by Yndestad et al [84]. Notably, Yndestad et al. included PAH patients with varying etiological causes, including those caused by HIV, connective tissue disease, and liver cirrhosis. The INHBA was identified as highly methylated in PAECs from IPAH and HPAH patients in a study by Hautefort et al [96]. Studies with a larger number of PAH patients with different etiology are needed in order to strengthen current evidence on the involvement of activin A in PAH.

The most pressing question is what causes the upregulation of activin A in PAH. Activin A expression may be triggered by hypoxia, as previously demonstrated in isolated human lung ECs [85]. Merfeld-Clauss et al. found that hypoxia could also increase activin A in ECs under normal physiological conditions [87]. The hypoxia-induced increase could be prevented by silencing hypoxia-responsive factors HIF1α and HIF2α, raising the possibility that activin A may be upregulated via HIF pathways in PAH. Moreover, aberrant HIF signaling is a major contributor to vascular remodeling in PAH. Therefore, it would be interesting to determine whether the HIF-activin axis contributes to PAH development [9]. The signaling upstream of activin A should also be elucidated and might provide therapeutic targets.

In addition to the aforementioned HIF-related hypoxia signaling, activin A is also known to affect vascular endothelial cell growth factor (VEGF) pathways, although several reports have provided conflicting evidence on the matter [88,97]. Activin A has been reported to inhibit VEGF signaling, yet other reports suggest that activin A may also enhance it, depending on the context. Following activin A treatment, an increase in endothelin-1 (ET-1) and plasminogen activator inhibitor-1 (PAI-1) have been reported in PASMCs. We also observed an increase in interferon-β (IFN-β) mRNA expression in PAECs under INHBA overexpression [84,85]. ET-1 is established as a factor that promotes PAH development while the involvement of PAI-1 and IFN-β in PAH has been previously reported. This suggests that activin A may affect these pathways in addition to the ones previously discussed [98,99,100]. Further research is necessary to determine the relationship between these molecules and activin A. In addition, considering the fact that activins and inhibins are glycoproteins, it is conceivable that the effects of activin A could extend beyond the signaling already discussed. Various forms of glycoproteins have been implicated in the development of PAH, where these could affect the metabolic phenotype of vascular cells, which could in turn contribute to the pro-proliferative and apoptotic phenotype seen in the pulmonary vasculature of PAH patients [101]. Lastly, considering that activin A could play a role in macrophage activation similar to that induced via LPS, the molecular mechanisms that bridge it with pro-inflammatory signaling during PAH warrant further discussion.

In contrast to activin A, other members of the activin/inhibin family have not been extensively studied under PH conditions, and much less in the context of PAH. It was reported that the activin/inhibin homolog of *S. mansoni*, causing schistosomiasis-related PAH, was highly expressed [102]. Activin B was noted to induce the expression of myogenic and fibrogenic genes in PASMCs in addition to the pro-inflammatory activation of macrophages, contributing to PAH development and progression [86,94]. Therefore, activin B may play a similar role to activin A in PAH development, which requires further research. Both activin A and B are known as adipokines implicated in metabolic syndrome, a condition found among PAH patients. Thus, there might be a connection between these comorbidities and activins/inhibins, which requires further study. The effects of activin A can be counteracted by inhibin A and follistatin. The expression of INHA, however, did not change significantly even after the overexpression of INHBA, suggesting that PAECs do not display a compensatory increase in INHA expression [85]. In contrast, a report stated that follistatin mRNA expression levels increased in the PASMCs of mice. This result was not observed in the PAECs under INHBA overexpression [84]. Lastly, although it is beyond the scope of the current manuscript, we should note that many studies are now trying to confirm the role of activin A in conditions related to PH groups II and III. It is clear that activin/inhibin, particularly activin A, plays an important role in PAH development. Therefore, proteins of this family represent promising therapeutic targets.

## 5. Clinical Landscape of Activin-Targeting Treatments in PAH

There has been great progress in the development of drugs targeting activin/inhibin over the past several years, with a growing focus on the additional “tools” for combating dysregulated activin signaling in light of increasing evidence on its physiological and pathophysiological relevance. As the most extensively studied family member, activin A has been the major focus point in related treatment strategies [103]. The agents currently known to target activin A are listed in Table 3. Sotatercept works as a ligand trap mimicking activin receptor type IIA (ActRIIA-Fc) that binds not only activin A, but also GDF8 and GDF11, members of the growth differentiation factor family [104]. Several preclinical studies, demonstrating the efficacy of this drug in patients with PAH, were discussed in the previous segment.

Sotatercept was introduced as a first-in-class drug in 2022, having shown promising results in a phase II randomized clinical trial (PULSAR) that recruited patients with PAH due to various reasons (excluding those with diseases caused by HIV, schistosomiasis, and portal hypertension). Considering the relative heterogeneity of the patient cohort, the trial results were promising. More specifically, the administration of sotatercept at two different dosages (0.3 or 0.7 mg/kg body weight every 3 weeks), in addition to the PAH-specific treatment, resulted in significant improvements in the hemodynamic features of patients relative to the placebo, particularly PVR [93]. Furthermore, the effect of sotatercept was not only limited to hemodynamic improvement, but also led to an improvement in the 6-min walk distance and a decrease in serum N-terminal pro-BNP levels [93]. In addition, patients experienced an improvement in clinical symptoms with tolerable adverse events. Thrombocytopenia and increased hemoglobin levels were the most common adverse events. In the extension study of PULSAR, the 97 patients originally enrolled in PULSAR were re-enrolled, and those who were originally randomized to the placebo were re-randomized to receive either 0.3 or 0.7 mg/kg body weight every 3 weeks (placebo-crossed) compared to the group that continued the sotatercept treatment already received in the original trial [104]. Significant improvements in hemodynamic and clinical parameters were observed in the placebo-crossed groups. In the patients who continued sotatercept treatment, the treatment was effective for a prolonged period without significant adverse events occurring. This indicates an acceptable safety profile. Encouraged by the results of the PULSAR trial, several PAH trials are ongoing or have just been completed. These include the phase III STELLAR trial (NCT04576988) and the continuation of the PULSAR trial, which has just announced several promising hemodynamic and clinical results. In addition, the SOTERIA trial (NCT04796337) analyzes the effect of sotatercept in a long-term administration setting of up to 200 weeks. The HYPERION trial (NCTNCT04811092) examines the use of sotatercept in newly diagnosed intermediate- and high-risk patients with PAH. The ZENITH trial analyzes the use of sotatercept in PAH patients with WHO functional class III and IV (NCT04896008). In addition to sotatercept, RAP-011, another sotatercept analog, is reportedly effective in treating experimental PH in vivo and is currently being investigated for treating PAH conditions [94]. Furthermore, RKER-012, another ligand trap that targets a different activin receptor than sotatercept (ActRIIB) is currently being evaluated for amelioration of the PH phenotype in the experimental SU5416/Hypoxia rat model [105]. Taken together, sotatercept or other ligand trap-based therapeutic agents could prove to be the major breakthrough targeting a previously overlooked major pathway involved in PAH pathogenesis.

At present, the activin/inhibin family is actively investigated in the context of various other pathologies, including cancer and muscular dystrophy. All drugs with a mechanism of action similar to sotatercept, including STM-434, ramatercept, luspatercept, and bimagrumab, have been tested under aforementioned cancer and/or muscular dystrophy conditions mentioned above [106,107]. Moreover, garetosmab, an antibody against activin A, is being tested for progressive fibrodysplasia ossificans, and the results from phase II testing have been encouraging [107]. Despite not being tested in PAH treatment, these drugs appear to act as ligand traps for activin A. In non-PAH activin-associated conditions, alternative methods for drug delivery have been explored. The extracellular inhibitor of activin A, follistatin, is another relevant therapeutic in this context, acting through a different mechanism. Follistatin is also capable of inhibiting myostatin, a protein thought to be responsible for muscular dystrophy, with follistatin administration being of particular interest in this as means for enhancing follistatin levels in the human body [70]. One example is the phase I/IIA clinical trial for the use of a follistatin construct introduced via an adeno-associated virus for the treatment of Becker muscular dystrophy [108]. Recombinant follistatin has also been developed, such as ACE-083, although results on its efficacy against facioscapulohumeral muscular dystrophy conditions have not been promising [107,109]. Furthermore, because other isoforms of activin and inhibin have not been identified as relevant in PAH conditions, other agents that could restore the balance of activin/inhibin signaling have not been thoroughly investigated in clinical settings. In the future, more such agents will be tested as the involvement and roles of specific family members in PAH become increasingly clear.

## 6. Conclusions

It is now established that activins and inhibins play an important role in PAH pathogenesis. We believe that maximizing the therapeutic potential of activin/inhibin in PAH requires further in-depth basic and clinical research.

## Figures and Tables

**Figure 1 ijms-24-03332-f001:**
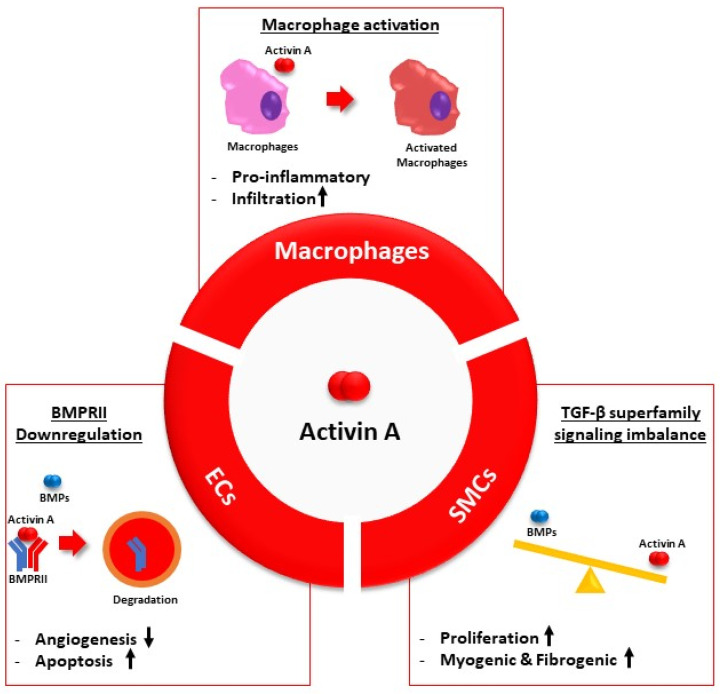
Summary of the activin A mechanism in PAH development. EC, endothelial cells. SMC, smooth muscle cells. BMPRII, bone morphogenetic protein receptor type 2. BMP, bone morphogenetic protein. TGF-β, transforming growth factor β. Arrow up, increase; arrow down, decrease.

**Table 1 ijms-24-03332-t001:** Clinical classification of pulmonary arterial hypertension.

Group 1 Pulmonary arterial hypertension	
1.1	Idiopathic
1.1.1	Non-responders at vasoreactivity testing
1.1.2	Acute responders at vasoreactivity testing
1.2	Heritable
1.3	Associated with drugs and toxins
1.4	Associated with:
1.4.1	Connective tissue disease
1.4.2	HIV infection
1.4.3	Portal hypertension
1.4.4	Congenital heart disease
1.4.5	Schistosomiasis
1.5	PAH with features of venous/capillary involvement
1.6	Persistent PH of the newborn
Group 2 PH associated with left heart disease	
2.1	Heart failure:
2.1.1	With preserved ejection fraction
2.1.2	With reduced or mildly reduced ejection fraction
2.2	Valvular heart disease
2.3	Congenital/acquired cardiovascular conditions leading to post-capillary PH
Group 3 PH associated with lung diseases and/or hypoxia	
3.1	Obstructive lung disease or emphysema
3.2	Restrictive lung disease
3.3	Lung disease with mixed restrictive/obstructive pattern
3.4	Hypoventilation syndromes
3.5	Hypoxia without lung disease (e.g., high altitude)
3.6	Developmental lung disorders
Group 4 PH associated with pulmonary artery obstructions	
4.1	Chronic thrombo-embolic PH
4.2	Other pulmonary artery obstructions
Group 5 PH with unclear and/or multifactorial mechanisms	
5.1	Hematological disorders
5.2	Systemic disorders
5.3	Metabolic disorders
5.4	Chronic renal failure with or without hemodialysis
5.5	Pulmonary tumor thrombotic microangiopathy
5.6	Fibrosing mediastinitis

**Table 2 ijms-24-03332-t002:** Known isoforms of activins and inhibins.

Isoform Name	Chains
Activin A	INHBA + INHBA
Activin B	INHBA + INHBB
Activin AB	INHBB + INHBB
Activin C	INHBA + INHBC
Activin E	INHBA + INHBE
Inhibin A	INHBA + INHA
Inhibin B	INHBB + INHA

**Table 3 ijms-24-03332-t003:** Current therapeutic options targeting activin A.

Agent Name	Mechanism of Action	Phase
Sotatercept	ActRIIA ligand-trap	Phase III trial ongoing
RAP-011	ActRIIA ligand-trap	Preclinical
RKER-012	ActRIIB ligand-trap	Preclinical

## Data Availability

Data are contained within this article.

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
