# Peer review of "Inactivating the Uninhibited: The Tale of Activins and Inhibins in Pulmonary Arterial Hypertension"

_ijms, 2023, doi:10.3390/ijms24043332_

Round 1

Reviewer 1 Report

this PAH condition, not specific lesions, so, see extending the introduction with declaring the causes, pathogenesis, and clinical observations associated with PAH conditions 

and add other references such as (Schermuly, R. T., Ghofrani, H. A., Wilkins, M. R., & Grimminger, F. (2011). Mechanisms of disease: pulmonary arterial hypertension. Nature Reviews Cardiology8(8), 443-455.)

you can write the relationship between bacterial LPS, Lipid profiles, and PAH condition 

Author Response

this PAH condition, not specific lesions, so, see extending the introduction with declaring the causes, pathogenesis, and clinical observations associated with PAH conditions 

Response: Thank you for your constructive comment. We have extended the introduction with regard to background on PAH (page 1).

and add other references such as (Schermuly, R. T., Ghofrani, H. A., Wilkins, M. R., & Grimminger, F. (2011). Mechanisms of disease: pulmonary arterial hypertension. Nature Reviews Cardiology8(8), 443-455.)

Response: Thank you for your comment. We have added the mentioned reference and cited it in our manuscript. (reference number 7).

you can write the relationship between bacterial LPS, Lipid profiles, and PAH condition 

Response: Thank you for your suggestion. We have included discussion on LPS and lipid profiles as well as their relation to PAH conditions on pages 3, 4, 8, and 10.

Reviewer 2 Report

Dear authors,

this is a well written manuscript on an underrepresented, but important topic.  

Anyhow, from my point of view I do not have major concerns and would recommend publication.

Author Response

Dear authors,

this is a well written manuscript on an underrepresented, but important topic.  

Anyhow, from my point of view I do not have major concerns and would recommend publication.

Response: Thank you your kind comments. We are encouraged by your positive review and hope that our manuscript will contribute to current knowledge on this topic.

Reviewer 3 Report

Dear Editor

Thank you very much for the opportunity to refer to the work titled: "Inactivating the Uninhibited...".

The publication has the character of a narrative review, the title itself reflects its journalistic rather than scientific character.

Beginning with the introduction, the authors downplay knowledge of the problem and the prevalence of PAH syndrome. They treat this immensely complex etiological and pathophysiological disease problem in a vastly oversimplified manner. Concern is introduced from the beginning by the given definition of the disease [41-43], where from the first sentences [25-26] the described problem is devoted to one of the disease entities of PAH- pre-capillary PAH, while the term used in the definition of PCWP size of at least 15 mm refers to extra-capillary PAH, associated with LV cardiac dysfunction. In this type of story, a prerequisite for credibility is the maintenance of a certain framework and standards, which we can find in the 2022 ESC/ERS recommendations [doi: 10.1093/eurheartj/ehac237. PMID: 36017548].

Subsequently, the authors describe, still without considering the division of the disease entity, selectively the pathomechanisms of PAH and the methodology of contemporarily accepted drug therapies like prostacyclin analogs.

Moving on to the substance of the story, we again get an immeasurably large amount of not entirely logically related information on the described actions Activin-A  a glycoprotein, which belongs to the transforming growth factor-β. Arguably, the involvement of glycoproteins in the apoptosis of cells their proliferation and angiogenesis in a very indirect way can be linked to the described problem of PAH.

Chapter 5, devoted to the use of Activin in the treatment of PAH, presents a Table 2 that should be a summary of ongoing clinical trials. This concept summary, however, does not present clinical trials devoted to the disease entity described, but clinical work in experimental therapy of cancer and genetic disorders is presented.

The tale of signaling proteins is unsystematic, not taking into account the standards of the recognized terminology of the PAH disease entity and its division. Referring to idiopathic PAH (IPAH) seriously confuses the definition of PAH with that in the course of LV cardiac dysfunction (IpcPH, CpcPH). The paper in its current form is not suitable for publication.

Author Response

Dear Editor

Thank you very much for the opportunity to refer to the work titled: "Inactivating the Uninhibited...".

The publication has the character of a narrative review, the title itself reflects its journalistic rather than scientific character.

Beginning with the introduction, the authors downplay knowledge of the problem and the prevalence of PAH syndrome. They treat this immensely complex etiological and pathophysiological disease problem in a vastly oversimplified manner. Concern is introduced from the beginning by the given definition of the disease [41-43], where from the first sentences [25-26] the described problem is devoted to one of the disease entities of PAH- pre-capillary PAH, while the term used in the definition of PCWP size of at least 15 mm refers to extra-capillary PAH, associated with LV cardiac dysfunction. In this type of story, a prerequisite for credibility is the maintenance of a certain framework and standards, which we can find in the 2022 ESC/ERS recommendations [doi: 10.1093/eurheartj/ehac237. PMID: 36017548].

Response: Thank you for your correction. We have revised the definition as advised, on page 2 line 43-54.

Subsequently, the authors describe, still without considering the division of the disease entity, selectively the pathomechanisms of PAH and the methodology of contemporarily accepted drug therapies like prostacyclin analogs.

Response: Thank you your suggestion. We have emphasized on the difference between PAH types throughout our manuscript, adding a new Table 1 on page 2, which provides a description of PAH nomenclature. We have also described the mechanism of action for other accepted PAH drugs on page 4 line 102-116.

Moving on to the substance of the story, we again get an immeasurably large amount of not entirely logically related information on the described actions Activin-A  a glycoprotein, which belongs to the transforming growth factor-β. Arguably, the involvement of glycoproteins in the apoptosis of cells their proliferation and angiogenesis in a very indirect way can be linked to the described problem of PAH.

Response: We have restructured the information on Activin A to give a better description of its role in PAH development in section 3. Furthermore, we also address the matter of activins and inhibins being glycoproteins on page 10 line 388-393.

Chapter 5, devoted to the use of Activin in the treatment of PAH, presents a Table 2 that should be a summary of ongoing clinical trials. This concept summary, however, does not present clinical trials devoted to the disease entity described, but clinical work in experimental therapy of cancer and genetic disorders is presented.

Response: We have removed clinical trials unrelated to PAH from Table 2 (now revised as Table 3).

The tale of signaling proteins is unsystematic, not taking into account the standards of the recognized terminology of the PAH disease entity and its division. Referring to idiopathic PAH (IPAH) seriously confuses the definition of PAH with that in the course of LV cardiac dysfunction (IpcPH, CpcPH). The paper in its current form is not suitable for publication.

Response: We have provided PAH nomenclature in order to avoid confusion and thus hope that our revised manuscript is now suitable for publication.

Round 2

Reviewer 3 Report

Thank you for all the changes made to the manuscript. The publication shows the current state of knowledge on the PAH problem. It also points out new therapeutic concepts of the disease. The current version of the paper is suitable for publication.